# Effect of Psilocybin and Ketamine on Brain Neurotransmitters, Glutamate Receptors, DNA and Rat Behavior

**DOI:** 10.3390/ijms23126713

**Published:** 2022-06-16

**Authors:** Adam Wojtas, Agnieszka Bysiek, Agnieszka Wawrzczak-Bargiela, Zuzanna Szych, Iwona Majcher-Maślanka, Monika Herian, Marzena Maćkowiak, Krystyna Gołembiowska

**Affiliations:** 1Department of Pharmacology, Unit II, Maj Institute of Pharmacology, Polish Academy of Sciences, Smętna 12, 31-343 Kraków, Poland; wojtas@if-pan.krakow.pl (A.W.); bysiek@if-pan.krakow.pl (A.B.); szych@if-pan.krakow.pl (Z.S.); herian.monika@gmail.com (M.H.); 2Department of Pharmacology, Laboratory of Pharmacology and Brain Biostructure, Maj Institute of Pharmacology, Polish Academy of Sciences, Smętna 12, 31-343 Kraków, Poland; bargiela@if-pan.krakow.pl (A.W.-B.); majcher@if-pan.krakow.pl (I.M.-M.); mackow@if-pan.krakow.pl (M.M.)

**Keywords:** dopamine, serotonin, glutamate, GABA, microdialysis, DNA damage, glutamate receptors, light–dark box test, open field test, forced swim test

## Abstract

Clinical studies provide evidence that ketamine and psilocybin could be used as fast-acting antidepressants, though their mechanisms and toxicity are still not fully understood. To address this issue, we have examined the effect of a single administration of ketamine and psilocybin on the extracellular levels of neurotransmitters in the rat frontal cortex and reticular nucleus of the thalamus using microdialysis. The genotoxic effect and density of glutamate receptor proteins was measured with comet assay and Western blot, respectively. An open field test, light–dark box test and forced swim test were conducted to examine rat behavior 24 h after drug administration. Ketamine (10 mg/kg) and psilocybin (2 and 10 mg/kg) increased dopamine, serotonin, glutamate and GABA extracellular levels in the frontal cortex, while psilocybin also increased GABA in the reticular nucleus of the thalamus. Oxidative DNA damage due to psilocybin was observed in the frontal cortex and from both drugs in the hippocampus. NR2A subunit levels were increased after psilocybin (10 mg/kg). Behavioral tests showed no antidepressant or anxiolytic effects, and only ketamine suppressed rat locomotor activity. The observed changes in neurotransmission might lead to genotoxicity and increased NR2A levels, while not markedly affecting animal behavior.

## 1. Introduction

Mood and anxiety disorders are among the leading causes of disability worldwide and place an enormous burden on society [1]. Major depressive disorder affects up to 5% of adults worldwide, leading to an increased susceptibility to a plethora of both mental and somatic diseases [2], and more importantly, an increased risk of suicide [3]. Currently used antidepressant drugs exert an effect after weeks or even months of treatment, and are not effective in so-called “treatment-resistant” depression [4]. Recently, it has been demonstrated that some dissociative drugs may alleviate the symptoms of major depressive disorder only hours after administration, and are also effective in treatment-resistant patients [5,6].

Studies conducted during the last 30 years show that ketamine exhibits strong antidepressant properties while working extremely fast after its administration [7,8]. Ketamine is a non-competitive NMDA receptor antagonist, used primarily as an anesthetic, but it can rapidly alleviate depressive symptoms after a single injection when used at subanesthetic doses [9]. Sadly, the effects only last for about 2 weeks; moreover, the drug has a plethora of side effects, and can be abused [7]. Ketamine’s antidepressant effect results from a burst of glutamate, which stimulates the mTOR pathway, leading to enhanced synaptic plasticity, which is disrupted in depressive patients [4]. This phenomenon translates into increased synaptic density in the frontal cortex and hippocampus [10,11].

Psychedelics might be some of the oldest psychoactive substances known to humankind [12]. They can be divided into two main categories by structure: indoleamines, e.g., DMT (N,N-dimethyltryptamine), psilocybin or LSD (Lysergic acid diethylamide), and phenylalkylamines, e.g., mescaline or DOI (2,5-Dimethoxy-4-iodoamphetamine) [13]. While the former demonstrate an affinity for several subtypes of 5-HT receptors [14], the latter bind mainly to the 5-HT2 receptor family [15,16]. Holistic data gathered from many studies indicate that psychedelics exert their psychoactive effects by activating cortical 5-HT2A receptors located on cortical pyramidal cells, leading to the release of glutamate [17,18,19,20,21], while possibly increasing thalamic GABA levels, resulting in increased sensory input to the cortex [22,23].

Similar to ketamine, psilocybin has been known to induce a rapid antidepressant effect that lasts for up to 6 months after a single administration, a level of effectiveness that is at least as high as “classic” antidepressants [6,24]. Those properties, such as in ketamine, are probably also mediated through elevated levels of glutamate stimulating the mTOR pathway [25]. Psychedelics enhance synaptic plasticity to a comparable or even greater degree than ketamine, promoting neuro- and synaptogenesis [25]. At the same time, they are non-addictive and seem to exhibit fewer side effects than ketamine [26].

The mechanisms underlying the antidepressant properties of both ketamine and psilocybin are still not fully understood. Detailed comparison studies should be conducted to evaluate their mutual properties. In addition, their safety and long-lasting effects should be carefully explored if they are to be used as a standard treatment for mood disorders, not only as a curiosity applied in experimental therapies. Psychedelics acting at 5-HT2A receptors in the reticular nucleus of thalamus can disrupt sensory input to the cortex and alter pyramidal cells’ signaling [27]. Possible psilocybin-induced serotonin syndrome is of legitimate concern, as 5-HT2 receptor agonist DOI-treatment in wild mice resulted in oxidative stress and mitochondrial impairment, which led to cellular oxidative damage [28]. To address those questions, we have examined the effect of a single administration of both ketamine and psilocybin on the extracellular levels of dopamine (DA), serotonin (5-HT), glutamate and γ-aminobutyric acid (GABA) in the rat frontal cortex and reticular nucleus of the thalamus using in vivo microdialysis. The potential genotoxic effect in the frontal cortex and hippocampus was measured with the comet assay. MDMA was used in this test as a comparator as it was shown to augment DA and 5-HT release, which may be a source of oxidative stress [29]. Moreover, Western blot analysis was performed to measure the effect of the investigated drugs on selected protein levels in the rat frontal cortex. Finally, an open field test, light–dark box test and forced swim test were conducted to evaluate the effect on selected aspects of rat behavior. To distinguish long-lasting from acute, psychoactive effects, all the experiments (with the exception of microdialysis) were performed at least 24 h after administration of the chosen drug. The effect of psilocybin and ketamine on neurotransmitters’ release was never studied in detail, but by a direct comparison of their impact on the neuronal network the mechanism of psychedelics in the brain may be explained. An identification of the possible undesirable side effects of psilocybin and ketamine, such as genotoxicity, behavioral disturbances and possible adaptive changes in glutamate receptors, is of great importance. Our findings may be helpful in the definition of the therapeutic capabilities of psychedelics.

## 2. Results

### 2.1. Effect of Psilocybin and Ketamine on Extracellular Levels of DA, 5-HT, Glutamate and GABA in the Rat Frontal Cortex

Psilocybin at a dose of 2 mg/kg, but not at 10 mg/kg, significantly increased extracellular levels of DA up to ca. 200% of the baseline in the rat frontal cortex (Figure 1A). Ketamine (10 mg/kg) was very potent in increasing (up to ca. 500% of the baseline) the DA extracellular level (Figure 1A). Repeated-measures ANOVA showed the significant effect of treatment groups (F_3,39_ = 192, *p* < 0.0001), sampling period (F_11,429_ = 67, *p* < 0.0001) and the interaction between treatment groups and sampling period (F_33,429_ = 36, *p* < 0.0001). Total effects expressed as AUC, shown in Figure 1B, were significantly increased for 2 mg/kg psilocybin and ketamine (F_3,39_ = 192, *p* < 0.001, one-way ANOVA).

The extracellular 5-HT level was increased in the rat frontal cortex by both doses of psilocybin (up to 200–250% of the baseline) and more potently by ketamine (up to 350% of the baseline) (Figure 1C). Repeated-measures ANOVA showed the significant effect of treatment groups (F_3,29_ = 537, *p* < 0.0001), sampling period (F_11,319_ = 71, *p* < 0.0001) and the interaction between treatment groups and sampling period (F_33,319_ = 38, *p* < 0.0001). The total effects expressed as AUC, shown in Figure 1D, were significantly increased for both psilocybin doses, 2 and 10 mg/kg, and for ketamine (F_3,29_ = 537, *p* < 0.0001, one-way ANOVA).

The extracellular glutamate level was slightly but significantly decreased by the lower psilocybin dose of 2 mg/kg (to ca. 80% of the baseline), and markedly increased by psilocybin at the dose of 10 mg/kg (up to ca. 300% of the baseline) as well as by ketamine (up to 150% of the baseline) (Figure 1E). Repeated-measures ANOVA showed the significant effect of treatment groups (F_3,33_ = 326, *p* < 0.0001), sampling period (F_11,3163_= 29, *p* < 0.0001) and the interaction between treatment groups and sampling period (F_33,363_ = 46, *p* < 0.0001). Total effects expressed as AUC, shown in Figure 1F, were significantly decreased by 2 mg/kg psilocybin and significantly increased by psilocybin 10 mg/kg and ketamine (F_3,33_ = 327, *p* < 0.0001, one-way ANOVA) (Figure 1F).

The extracellular level of GABA was slightly but significantly increased by the psilocybin dose of 2 mg/kg (up to ca. 120% of the baseline), more potently by the higher dose of 10 mg/kg (up to ca. 200% of baseline) and by ketamine (up to 180% of baseline) (Figure 1G). Repeated measures ANOVA showed the significant effect of treatment groups (F_3,35_ = 277, *p* < 0.0001), sampling period (F_11,385_ = 14, *p* < 0.0001), and the interaction between treatment groups and sampling period (F_33,385_ = 29, *p* < 0.0001). Total effects expressed as AUC, shown in Figure 1H, were significantly increased by both psilocybin doses and ketamine (F_3,35_ = 276, *p* < 0.0001, one-way ANOVA).

Basal levels of DA, 5-HT, glutamate and GABA are shown in Appendix A.

### 2.2. Effect of Psilocybin and Ketamine on Extracellular Levels of Glutamate and GABA in the Rat Reticular Nucleus of the Thalamus

Psilocybin and ketamine had no effect on the extracellular level of glutamate in the rat reticular thalamus nucleus (Figure 2A). Repeated-measures ANOVA did not show the significant effect of treatment groups (F_3,23_ = 1,31, *p* < 0.29), but there was a significant effect for sampling period (F_11,253_ = 12, *p* < 0.0001) and the interaction between treatment groups and sampling period (F_33,253_ = 4.1, *p* < 0.0001). Total effects expressed as AUC, shown in Figure 2B, were not changed by psilocybin doses or ketamine (F_3,23_ = 1.29, *p* < 0.30, one-way ANOVA).

The extracellular level of GABA was slightly but significantly increased by psilocybin at a low dose; however, it was potently increased by psilocybin at a dose of 10 mg/kg (up to ca. 170% of the baseline) (Figure 2C). Ketamine did not affect GABA extracellular level. Repeated-measures ANOVA showed the significant effect of treatment groups (F_3,22_ = 22, *p* < 0.0001). There was no significant effect for sampling period (F_11,242_ = 1.73, *p* < 0.07), but there was a significant interaction between treatment groups and sampling period (F_33,242_ = 3.11, *p* < 0.0001). Total effects expressed as AUC, shown in Figure 2D, were significantly increased by both doses of psilocybin but not by ketamine (F_3,22_ = 33, *p* < 0.0001, one-way ANOVA).

Basal levels of glutamate and GABA are shown in Appendix A.

### 2.3. The NMDA and AMPA Receptor Subunit Level in the Rat Frontal Cortex

The higher dose of psilocybin (10 mg/kg) significantly increased the GluN2A protein level by ca. 40% over the control, measured 24 h after administration (F_3,44_ = 9.3, *p* < 0.0001, one-way ANOVA) (Figure 3A,B). In contrast, ketamine administration decreased the GluN2A protein level by ca. 30%, but this effect was not statistically significant (*p* < 0.16 vs. control group). GluN2B protein levels were not changed by psilocybin or ketamine administration (F_3,20_ = 2.36, *p* < 0.10) (Figure 3A,C).

The protein levels of two AMPA receptor subunits, GluA1 and GluA2, measured 24 h after administration of psilocybin at doses of 2 and 10 mg/kg or ketamine at a dose of 10 mg/kg were not changed (F_3,20_ = 0.22, *p* < 0.88 and F_3,20_ = 0.85, *p* < 0.48, respectively; one-way ANOVA) (Figure 3D–F).

The uncropped, untouched, full original images of Western blots are shown in Appendix A.

### 2.4. The Effect of Psilocybin and Ketamine on DNA Damage in the Rat Frontal Cortex and Hippocampus

Psilocybin at a dose of 2 mg/kg did not produce DNA damage by reactive oxygen species (ROS) in the frontal cortex and hippocampus, presented as a percentage of tail moment 7 days after administration. However, psilocybin at a higher dose of 10 mg/kg significantly increased DNA damage in both brain regions (Figure 4A,B). Ketamine (10 mg/kg) produced DNA damage only in the hippocampus, while MDMA, used as a comparative compound, caused potent damage of DNA in the frontal cortex and hippocampus. One-way ANOVA showed the significant effect of treatment in the frontal cortex (F_4,26_ = 39, *p* < 0.0001) and hippocampus (F_4,26_ = 24, *p* < 0.0001).

### 2.5. The Effect of Psilocybin and Ketamine on Rat Behavior in the Open Field, Light–Dark Box and Forced Swimming Tests 24 h after Administration

Psilocybin did not affect time of walking and episodes of crossing in the open field test; however, ketamine decreased both parameters in this test (F_3,35_ = 39, *p* < 0.0001, F_3,35_ = 6.4, *p* < 0.01, respectively; one-way ANOVA) (Figure 5A,B).

The time spent in the dark compartment was longer than in the light zone for all groups of animals estimated in the light–dark test, but any difference was observed between treatments (Figure 5D). Two-way ANOVA showed no effect of treatment (F_3,72_ = 0, *p* < 1.0), a significant difference between the dark and light zones (F_1,72_ = 919, *p* < 0.0001) and no significant interaction between treatment and the light/dark zone (F_3,72_ = 1.17, *p* < 0.32). Exploration of the dark zone expressed as ambulatory distance was significantly increased in comparison to exploration of the light zone for all experimental groups (Figure 5C). Psilocybin at a dose of 2 mg/kg and ketamine significantly decreased exploration of the dark and light zones. Two-way ANOVA showed the effect of treatment (F_3,72_ = 17.4, *p* < 0.0001), a significant difference between the dark and light zone (F_1,72_ = 412, *p* < 0.0001), and no significant interaction between treatment and the light/dark zone (F_3,72_ = 1.73, *p* < 0.16).

Psilocybin at a dose of 2 mg/kg significantly increased immobility and swimming time (Figure 6A,C), while significantly decreasing climbing time (Figure 6B). Ketamine only significantly increased swimming time (Figure 6C). One-way ANOVA showed the significant effect of treatment on immobility time (F_3,76_ = 8.12, *p* < 0.0001), climbing time (F_3,76_ = 15, *p* < 0.0001) and swimming time (F_3,76_ = 27, *p* < 0.0001).

## 3. Discussion

Current studies suggest that both ketamine and psychedelics share rapid antidepressant properties, indicating the existence of convergence between their mechanisms of action [25,30]. The gathered data strongly supports the important role of glutamate in the frontal cortex [4]. Ketamine blocks NMDA receptors located on cortical GABAergic interneurons, leading to the disinhibition of glutamatergic neurotransmission and resulting in increased glutamate release [31], while psychedelics activate 5-HT2A receptors located on pyramidal neurons, stimulating the release of glutamate [20].

Our microdialysis experiments reinforce this hypothesis, showing a significant elevation of extracellular glutamate in rat frontal cortex levels after administration of either ketamine (10 mg/kg) or a high dose of psilocybin (10 mg/kg). Moreover, both drugs strongly affected DA, 5-HT and GABA levels, though the exact mechanisms responsible for this phenomenon may differ between the substances.

The data suggest that low doses of ketamine used for clinical and preclinical studies can block NMDA receptors on subsets of GABA interneurons and reduce interneuron firing prior to increasing pyramidal neuron firing [32]. As cortical GABAergic interneurons seem to be the primary target involved in ketamine’s antidepressant effect [33], a decrease in the extracellular levels of GABA should be expected. However, we observe an increase in GABA in parallel to an increase in the glutamate extracellular level in the frontal cortex. Our results are consistent with the findings of Pham et al., (2020) [34], who also found a concomitant increase in the release of both neurotransmitters in the medial prefrontal cortex of mice by local ketamine injection, but the effect was observed 24 h after the injection. NMDA receptors are composed of tetraheteromeric assemblies containing various combinations of GluN1-3 subunits and it has been shown that the subunit composition of pre-, post- or perisynaptic locations is not the same [35]. GluN2D subunits, highly expressed in parvalbumin-expressing GABAergic interneurons, may be involved in the cortical disinhibition by ketamine to a greater extent, but other subsets of interneurons may be blocked less effectively by ketamine, thus resulting in increased extracellular GABA. Another hypothesis of Duman et al. (2019) [36] suggests that the enhancement of GABAergic neurotransmission is caused by the initial burst of glutamate. Ketamine disturbs tonic inhibition of pyramidal cells and increases the burst of glutamate, leading to the release of both neurotransmitters.

In our study, ketamine markedly increased the extracellular 5-HT level, indicating the association of serotonin transmission with glutamate. A microdialysis study in monkeys showed that subanesthetic doses of ketamine transiently increased 5-HT extracellular levels in the prefrontal cortex by inhibiting SERT activity [37]. In line, the depletion of 5-HT by para-chlorophenylalanine attenuated the acute antidepressant effect of ketamine, suggesting that endogenous 5-HT is partly involved in this effect [38]. In contrast, an increase in the extracellular 5-HT level by ketamine in the medial prefrontal cortex of mice was eliminated by the AMPA receptor antagonist NBQX given into dorsal raphe nuclei (DRN) [34]. Thus, ketamine seems to indirectly elevate 5-HT extracellular levels through the stimulation of AMPA receptors located in DRN. AMPA receptors might be also involved in regulating ketamine-induced DA release in the prefrontal cortex as the AMPA receptor antagonist NBQX blocks the effects of ketamine on DA release [39]. In our study, acute administration of ketamine potently increased the DA extracellular level in the frontal cortex of rats. The mechanism of ketamine’s action on the DA system is not fully established. The medial prefrontal cortex sends projections to the ventral tegmental area (VTA) [40]. Exposure to stress disrupts the activity of a circuit formed by DA and glutamate projections that connect the medial prefrontal cortex and VTA, while ketamine restores these circuits [41]. The meta-analysis study by Kokkinou et al. (2018) [42] presents an elevation of frontal cortical DA levels shown in animal subjects and healthy humans after treatment with ketamine.

Serotonergic psychedelics acting primarily on serotonergic receptors mimic serotonergic input by presynaptic facilitation of glutamatergic neurons in the sensory, motor and limbic cortices. Psilocybin, after entering the body, is rapidly converted in the liver into psilocin [43]. Psilocin has an affinity for many 5-HT receptor subtypes, but mainly 5-HT2A and 5-HT1A receptors play a potential role in mediating the actions of serotonergic psychedelics [30]. 5-HT2A and 5-HT1A receptors showed 80% co-expression in the same pyramidal neurons in the rat frontal cortex [44]. 5-HT2A agonism leads to increased membrane excitability, while the effect of 5-HT1A agonism is a decrease in membrane excitability [30]. The subpopulation of pyramidal neurons that have high levels of 5-HT2A receptors may exhibit excitation, whereas other neurons with a high expression of 5-HT1A receptors display inhibition. Thus, opposing receptor actions correspond to different pharmacological responses. A minor fraction of 5-HT1A and 5-HT2A receptors reside in GABAergic interneurons; there are also 5-HT2A receptors in thalamocortical axons in the frontal cortex [45]. In our study, a low dose of psilocybin had a very weak effect on glutamate and GABA extracellular levels; however, a higher dose increased the release of both neurotransmitters, but this effect was significant in the second phase of the experiment. It seems likely that glutamate release by psilocybin may originate in a different neuronal location, i.e., glutamate from thalamocortical nerve terminals and GABA from cortical interneurons; both effects are mediated via 5-HT2A receptors. Support for our results can be found in the study by Mason et al. (2020) [46], who demonstrated an increase in glutamate and GABA levels due to psilocybin in the human medial prefrontal cortex.

The DA extracellular level was only elevated to a certain extent by the low psilocybin dose. It was also limited in time, and appeared unrelated to the glutamate and GABA extracellular level. This data is difficult to explain on the basis of glutamatergic or GABAergic projections into midbrain DA neurons. Psilocybin displays a high and comparable affinity for 5-HT2A and 5-HT1A serotonin receptors [14]. The inhibitory action of 5-HT1A receptors expressed by GABAergic neurons disinhibits pyramidal cells. Active pyramidal cells stimulate neurons in VTA, resulting in an increased DA release in the frontal cortex. Alternatively, psilocybin-activated 5-HT2A receptors located in pyramidal cells may be responsible for an increase in cortical DA release [47]. Thus, the stimulation of descending excitatory projection by 5-HT1A or 5-HT2A receptors could enhance glutamate release in VTA, subsequently stimulating glutamate receptors and increasing DA release in the frontal cortex. However, this process depends on the dose of psilocybin. 

In our study, psilocybin at both doses increased the 5-HT extracellular level in the rat frontal cortex. The selective activation of 5-HT2A receptors enhances the release of glutamate, which acts on pyramidal AMPA receptors, stimulating a descending projection to DRN, thus resulting in increased 5-HT release in the frontal cortex. In turn, the activation of inhibitory 5-HT1A receptors on pyramidal neurons [44] counteracts the stimulatory effect of 5-HT2A receptors and decreases 5-HT release [48]. The same suppressing effect on 5-HT neurons can be mediated by 5-HT2A receptors located in GABAergic interneurons in DRN [49]. In our work, psilocybin’s effect on the cortical 5-HT extracellular level may result from the activation of both subtypes of 5-HT receptors: the activation of 5-HT2A/5-HT1A receptors in the pyramidal location resulting in the stimulation of descending projection to DRN. A relatively small difference between the effects of both doses may be related to a difference in the activation of excitatory 5-HT2A or inhibitory 5-HT1A receptors by psilocybin. An inverted “U” shape dose–response effect on 5-HT release in the frontal cortex was demonstrated by us for 25I-NBOMe [50]. It should be noted that Sakashita et al. (2015) showed a slight but significant increase in 5-HT extracellular level in the medial prefrontal cortex due to psilocin (10 mg/kg) [51].

Due to the localization of both NMDA receptors [52] and 5-HT2A receptors [53] on the GABAergic interneurons in the reticular nucleus of the thalamus, it was of particular interest to evaluate the effects of both tested substances on the extracellular levels of amino acids. The reticular nucleus is made of GABAergic interneurons and is thought to be the center of the negative-feedback loop in the thalamus, as it is innervated by other thalamic nuclei while it projects inhibitory inputs back into the thalamus [54], regulating the amount of information sent into the cortex [22]. The NMDA antagonists seem to inhibit its activity, leading to the disinhibition of thalamic activity [52], while psychedelics are hypothesized to stimulate it, leading to the attenuation of thalamic gating [23,27,55]. Surprisingly, we observed only half of the expected outcome: while psilocybin dose-dependently increased extracellular GABA levels, ketamine did not affect either glutamatergic or GABAergic neurotransmission; this phenomenon is hard to explain, and further studies should be conducted to address this issue.

The main mechanism of ketamine action is a blockade of NMDA receptors localized on interneurons that leads to the disinhibition of glutamatergic neurons. Ketamine and serotonergic psychedelics acting at 5-HT2A receptors increase glutamate release, which stimulates mTOR signaling through the activation of AMPA receptors. As was shown further, the antidepressant effects of ketamine are mediated by the GluN2B subunit located on GABAergic interneurons, but not by GluN2A subunits in glutamatergic neurons [33]. Ketamine-like effects were demonstrated for selective NR2B antagonists in rodent models [56] and in a clinical study showing a rapid antidepressant response by a selective NR2B receptor antagonist, ifenprodil [57]. Knockdown of the GluN2A subunit in the medial prefrontal cortex in mice did not block the antidepressant effect of the positive allosteric modulator of the NMDA receptor rapastinel [58]. However, a significant increase in the GluN2A subunit was observed by us 24 h after a high dose of psilocybin administration in the rat frontal cortex. The GluN2B subunit level showed only a tendency to increase, while neither subunit levels were significantly decreased by ketamine. A decrease in the expression level of the NR2B subunit was evidenced in the somatosensory cortex of mice at 24 and 72 h after administration of ketamine [59]. In contrast, the expression of *Nr2a* and *Nr2b* genes increased four weeks after cessation of treatment with LSD in the rat prefrontal cortex [60]. No effect was observed in our study 24 h after drug administration in the synaptic AMPA receptor subunits GluA1 and GluA2. It cannot be excluded that the increase in the level of NMDA receptor subunits 24 h after psilocybin administration might be related to the simultaneously observed enhancement in dendritic spine density induced by 5-HT2A receptor activation [61].

Compounds such as ketamine and serotonin psychedelics that are capable of promoting rapid plasticity have recently been defined as psychoplastogens. These substances are considered safe and not addictive. However, by producing an excessive glutamate release, they may induce excitotoxicity resulting in oxidative stress and neuronal atrophy. In our study, an increase in oxidative DNA damage was observed in the frontal cortex seven days after the administration of a single high dose of psilocybin but not ketamine. However, the appearance of double- and single-strand DNA breaks was observed in the hippocampus after treatment with a high dose of psilocybin and ketamine. These data suggest that potential risks may be associated with higher doses of psilocybin as well as ketamine. Importantly, in the recent study of Shin et al. (2021) [28], oxidative stress induced by amphetamine analog and 5-HT2A receptor agonist DOI contributes to DOI-induced serotonergic behaviors and cellular damage [28].

To discern between the acute, psychoactive effect induced by both substances and the persistent, antidepressant effect, our behavioral experiments were conducted 24 h post administration to limit high concentrations of circulating drugs and metabolites in the system. Ketamine negatively affected locomotor behavior in the open field test, while psilocybin had no effect. Moreover, ketamine and a low dose of psilocybin shortened the distance traveled in both the light and dark zone of the LDB apparatus, though a high dose of psilocybin had no effect; similarly, there was no effect on anxiety due to either substance. Acutely administered NMDA antagonists induce hyperlocomotion in rodents, while, for psychedelics having an inverted U-shaped response, low doses stimulate locomotor activity and high doses inhibit it [62]. We hypothesize that the observed changes in behavior might result from adaptive changes triggered to cope with the acutely induced effects of both drugs. Few studies have been conducted to assess the effect of either psilocybin or ketamine on behavior 24 h after injection and there is a clear lack of data. Surprisingly, none of the treatments reduced immobility time in the forced swim test. As suggested by Jefsen et al. (2019) [63], the forced swim test, while effective in evaluating the properties of more “traditional” compounds, might not be suitable for assessing the antidepressant effect of psychedelics. Moreover, our studies were conducted on naive animals; a recent article by Viktorov et al. (2022) [64] suggests that the effect of NMDA antagonists is limited when used on animals who are not subjected to any model of depression. On the other hand, our behavioral tests were performed 24 h after the administration of the drugs, and while that is enough to eliminate their acute effects, it might not be enough for the long-lasting effects to manifest fully, as Hibicke et al. (2020) [26] reported an antidepressant effect observed 7 days post psilocybin administration. In this context, a dissenting observation was reported in the latest study of Cao et al. 2022 [65], that the acute administration of LSD (30 min prior to the FST) significantly attenuated “depression-like” freezing behavior in the forced swimming test in naive mice. Taking into account the above studies, the time of test performance is critical for its outcome.

## 4. Materials and Methods

### 4.1. Animals

Adult male Wistar Han rats (280–350 g; Charles River, Sulzfeld, Germany) were used in all the experiments. The animals were initially acclimatized and housed (6 per cage) in environmentally controlled rooms (ambient temperature 23 ± 1 °C, humidity 55 ± 10% and 12:12 light:dark cycle). Rats were handled once daily before the beginning of the experiments; an enriched environment was not applied. The animals had free access to tap water and typical laboratory food (VRF 1, Special Diets Services, Witham, UK). All animal use procedures were conducted in strict accordance with European regulations for animal experimentation (EU Directive 2010/63/EU on the Protection of Animals Used for Scientific Purposes). The 2nd Local Institutional Animal Care and Use Committee (IACUC) in Kraków, Poland approved the experimental protocols for Experimentation on Animals (permit numbers: 112/2021, 324/2021 and 79/20226).

### 4.2. Drugs and Reagents

Ketamine hydrochloride was purchased from Tocris/Bio-Techne (Abingdon, UK) and psilocybin was synthesized at the Department of Medicinal Chemistry of the Maj Institute of Pharmacology using the method described by Shirota et al. (2003) [66]; both were dissolved in sterile water. All solutions were made fresh on the day of experiment. The dose of ketamine (10 mg/kg) was based on a report by Popik et al. (2022) [67], while doses of psilocybin (2 and 10 mg/kg) on work by Jefsen et al. (2019) [63]. MDMA purchased from Toronto Research Chemicals Inc. (Canada) was used as reference drug at the dose of 10 mg/kg. At this dose, it significantly and in a reliable manner induced oxidative DNA damage in the rat frontal cortex [29]. All drugs were given intraperitoneally (ip) in the volume of 2 mL/kg. Ketamine, xylazine hydrochlorides and sodium pentobarbital used for anesthetizing the animals came from Biowet Puławy (Puławy, Poland). All necessary chemicals of the highest purity used for analysis by high-performance liquid chromatography (HPLC) were obtained from Merck (Warszawa, Poland). O-phthalaldehyde (OPA), from Sigma-Aldrich (Poznań, Poland), was used for the derivatization of glutamate to an electroactive compound. The chemicals used for the alkaline comet assay were from Trevigen (Gaithersburg, MD, USA) and Merck (Warsaw, Poland). The reagents used in immunohistochemistry came from Sigma-Aldrich (Poznań, Poland), Vector Laboratories (Burlingame, CA, USA) and Proteintech (Manchester, UK). MDMA was purchased from Toronto Research Chemicals Inc. (Toronto, Canada). The control group was treated with 0.9% NaCl solution in the same way.

### 4.3. Brain Microdialysis

Ketamine and xylazine (75 and 10 mg/kg, respectively) were injected intramuscularly to anesthetize the animals. Microdialysis probes (MAB 4.15.3Cu and MAB 4.15.2Cu, AgnTho’s AB, Sweden) were implanted into the following brain structures using the determined coordinates (mm): frontal cortex AP +2.7, L +0.8, V −6.5, and reticular nucleus of thalamus AP −1.5, L +2.1, V −5.0 from the dura [68]. Seven days after implantation, probe inlets were connected to a syringe pump (BAS, West Lafayette, IN, USA) which delivered artificial cerebrospinal fluid composed of (mM) 147 NaCl, 4 KCl, 2.2 CaCl_2_ and 1.0 MgCl_2_ at a flow rate of 2 µL/min. The monitoring of extracellular levels of neurotransmitters has been performed in freely moving animals. Five baseline samples were collected every 20 min after the washout period of 2 h. The respective drugs were administered and dialysate fractions were collected for the next 240 min. As the experiment ended, the rats were terminated and their brains underwent histological examination to validate probe placement. Histological tracing of microdialysis probes in frontal cortex and thalamus are presented in Appendix A.

### 4.4. Extracellular Concentration of DA, 5-HT, Glutamate and GABA

Extracellular DA and 5-HT levels were analyzed using an Ultimate 3000 System (Dionex, Sunnyvale, CA, USA), electrochemical detector Coulochem III (model 5300; ESA, Chelmsford, MA, USA) with a 5020 guard cell, a 5040 amperometric cell and a Hypersil Gold C18 analytical column (3 μm, 100 × 3 mm; Thermo Fisher Scientific, Sunnyvale, CA, USA). The details of the method have been described elsewhere [29,69]. The chromatographic data were processed by the Chromeleon v.6.80 (Dionex, Sunnyvale, CA, USA) software package run on a personal computer. The limit of detection of DA and 5-HT in dialysates was 0.002 pg/10 μL for DA and 0.01 pg/10 μL for 5-HT.

Glutamate and GABA levels in the extracellular fluid were measured by HPLC with electrochemical detection after the derivatization of samples with OPA/sulfite reagent to form isoindole-sulfonate derivatives, as previously described [29,69]. The data were processed using Chromax 2005 (Pol-Lab, Warszawa, Poland) software on a personal computer. The limit of detection of glutamate and GABA in dialysates was 0.03 ng/10 μL and 6.4 pg/10 μL, respectively.

### 4.5. Alkaline Comet Assay

The alkaline comet assay was performed with the use of a CometAssay^®^ Reagent Kit for Single Cell Gel Electrophoresis. The animals were terminated by decapitation 7 days after drug injection and the frontal cortices and hippocampi were dissected. After homogenization and several stages of purification and centrifugation (as previously described in Wojtas et al. (2021)) [29], the nuclear suspension was obtained using a sucrose gradient (2.8 M/2.6 M, bottom to top). The nuclear fraction was mixed with low-melting point agarose and transferred immediately onto CometSlides™. The following steps, including membrane lysis, DNA unwinding, alkaline electrophoresis and staining (SYBR^®^ Gold), were carried out according to the Trevigen CometAssay^®^ protocol. Stained sections were acquired and analyzed under a fluorescence microscope (Nikon Eclipse50i, Nikon Corporation, Tokyo, Japan) equipped with a camera and NIS Elements software. The data was analyzed using OpenComet software v.1.3, a plugin of the ImageJ program v.1.47 (NIH, Bethesda, MD, USA). DNA damage was presented as a tail moment. The tail moment incorporates a measure of both the smallest detectable size of migrating DNA (reflected by the comet tail length) and the number of damaged pieces (represented by the intensity of DNA in the tail).

### 4.6. Western Blotting

The animals were terminated by decapitation and their brains were quickly removed from the skull. The frontal cortex was dissected and rapidly frozen in liquid nitrogen and stored at −20 °C. The Western blot procedure was performed as previously described in Maćkowiak et al. [70], 2019 and Latusz and Maćkowiak, 2020 [71]. The tissue was homogenized (TissueLyser, Retsch, Munich, Germany) in lysis buffer (PathScan^®^ Sandwich ELISA Lysis Buffer, Cell Signaling, Denver, CO, USA). Protein concentrations in the extracts were determined using a QuantiPro BCA Assay kit (Sigma-Aldrich, Poznań, Poland). The samples of equal protein content were adjusted to a final concentration of 10 mM Tris (pH 6.8) containing 2% SDS, 8% glycerol and 2% 2-mercaptoethanol with bromophenol blue as a marker and then boiled at 100 °C for 8 min. Protein extracts (10 μg of protein per lane, for GluN2A and GluA2 and 20 μg protein per lane for GluN2B and GluA1 analysis) were separated on 7.5% SDS-PAGE gel and transferred to nitrocellulose membranes using an electrophoretic transfer system (Bio-Rad Laboratories, Hercules, CA, USA); then, the membranes were stained with Ponceau S to confirm gel transfer. The membranes were then cut into three parts: the lower portion was used for GAPDH protein and the proteins with molecular weights greater than 37 kDa were determined from the next portions of the membrane. The blots were washed, and non-specific binding sites were blocked with 5% albumin (Bovine Serum Albumin; Sigma-Aldrich, Poznań, Poland) and blocking reagent (Lumi Light Western Blotting kit, Roche, Basil, Switzerland) in Tris-buffered saline (TBS) for 1 h at room temperature. Then, the blots were incubated overnight at 4 °C with the following primary antibodies: rabbit anti-GluN2A (1:500; AB1555P, Sigma-Aldrich, MercMillipore, Warszawa, Poland), rabbit anti-GluN2B (1:500; AB1557P, Sigma-Aldrich, MercMilipore, Warszawa, Poland), rabbit anti-GluA1 (1:1000; 04-855, Sigma-Aldrich, MercMilipore, Warszawa, Poland), rabbit anti-GluA2 (1:1000; AB1768-I, Sigma-Aldrich, MercMilipore, Warszawa, Poland) and rabbit anti-GAPDH (1:10,000; 14C10, 2118S Cell Signaling Technology, Denver, CO, USA). The peroxidase-conjugated secondary anti-rabbit IgG antibody (1:1000, Roche, Basil, Switzerland) was used to detect immune complexes (incubation for 1 h at room temperature). Blots were visualized using enhanced chemiluminescence (ECL, Lumi-LightPlus Western Blotting Kit, Roche, Basil, Switzerland) and scanned using a luminescent image analyzer (LAS-4000, Fujifilm, Boston, MA, USA). The molecular weights of immunoreactive bands were calculated on the basis of the migration of molecular weight markers (Bio-Rad Laboratories, Hercules, CA, USA) using Multi Gauge V3.0 (Fujifilm, Boston, MA, USA) software. The levels of analyzed proteins were normalized for GAPDH protein.

### 4.7. Open Field (OF) Test

The open field test was performed to modify the procedure described by Rogóż and Skuza (2011) [72]. A round black arena (1 m in diameter) was virtually divided into eight sections, creating a wheel. The test was conducted in a dimly lit room, with the middle of the arena illuminated by a 75 W light bulb placed at a height of 75 cm. Rats were placed in the middle of the arena 20 min after drugs injection. Their behavior was recorded for 10 min. The exploration was quantified with the following parameters: time of walking, number of line crossings, episodes of peeping under the arena, number of grooming events, and number of rearings.

### 4.8. Light–Dark Box (LDB) Test

This experimental procedure was performed in the TSE Fear Conditioning System (TSE System, Germany). The light/dark exploration test was performed as previously described by Chocyk et al. (2015) [73] and Bilecki (2021) [74]. Briefly, each experimental cage included an arena (45 × 45 × 45 cm) with a light compartment made of clear acrylic and a dark compartment made of black acrylic. The black compartment covered 33% of the total cage area, and the black dividing wall was equipped with a central tunnel gate (11 × 8.4 cm). The light compartment was brightly illuminated (100 lx), whereas the dark compartment received no light at all. The animals were kept in total darkness for 1 h prior to the testing, and the entire experiment was conducted with the room lights off. The animals were individually tested in single 10 min trials. At the beginning of each testing session, a rat was placed in the center of the light compartment, facing away from the gate. The behavioral responses during the test session were recorded using Fear Conditioning software (TSE, Bad Homburg, Germany). Specifically, the number of transitions between the compartments, the time spent in each compartment and locomotor activity (the distance traveled) were measured.

### 4.9. Forced Swim Test (FST) in Rats

On the first day of the FST (pre-test), the rats were placed individually in a cylinder (50 cm high × 23 cm in diameter) filled to a 30 cm depth with water (25 ± 1 °C) for 15 min, then removed from the water, dried with towels, placed in a warmer enclosure for 15 min and finally returned back to their home cages, as previously described (Detke et al., 1995) [75]. The cylinders were emptied and cleaned between rats. Twenty-four hours following the first exposure to forced swimming, the rats were retested for five minutes under identical conditions. Retest sessions were evaluated by two observers who were unaware of the treatment condition and who measured the swimming, climbing and immobility time. A rat was rated to be immobile if it was only making the necessary movements to keep its head above water; swimming behavior was defined as actively making swimming movements that caused the rat to move within the center of the cylinder or swim below the surface of water (diving); climbing behavior was recorded if a rat was making forceful thrashing movements with its forelimbs against the walls of the cylinder.

### 4.10. Statistical Analysis

Drug effects on DA, 5-HT, glutamate and GABA release in the brain regions were analyzed with repeated measures ANOVA on normalized responses followed by Tukey’s post hoc test. All obtained data were presented as a percentage of the basal level, assumed to be 100%. The data collected from the LDB test were analyzed with the two-way ANOVA followed by Tukey’s post hoc test. The results obtained in the open field test, forced swim test, Western blotting data and results obtained in the comet assay were analyzed with one-way ANOVA followed by Tukey’s post hoc test. The differences were considered significant if *p* < 0.05. The detected outliers were removed from the dataset using Grubb’s test. All statistical analyses were carried out using STATISTICA v.13.3 StatSoft Inc. 1984-2011 (TIBCO Software Inc., Palo Alto, CA, USA) and GraphPad Prism v.9.1.2 (GraphPad Software Inc., San Diego, CA, USA).

## 5. Conclusions

In conclusion, our results indicate that both psilocybin and ketamine exert a profound effect on thalamo-cortical neurotransmission. It seems likely that psilocybin and ketamine act on various molecular targets. Psilocybin activates 5-HT2A receptors; ketamine blocks subsets of NMDA receptors on GABA interneurons, which in turn disinhibits pyramidal cells. Both mechanisms result in the facilitation of glutamate release, which exerts stimulatory effect on dopaminergic VTA cells or serotonergic dorsal raphe neurons. As a consequence, this leads to an increase of DA and 5-HT levels. However, the modulatory role of 5-HT1A and 5-HT2C receptors in psilocybin’s effect cannot be excluded. Our findings also add neurochemical evidence that GABA neurons in the reticular nucleus of thalamus underlie the mechanism regulating the sensory information provided to the cortex by psilocybin. The increase in glutamate extracellular level in the frontal cortex after acute doses of psilocybin seems to correspond with changes in the NMDA receptor subunit GluN2A level. The DNA damage produced by higher doses of psilocybin and ketamine may result from glutamate-induced excitotoxicity and oxidative stress. The changes observed in the level of neurotransmitters do not translate into rat behavior tested 24 h after administration. No effect on anxiety and the reduction of immobility may result from adaptative mechanisms triggered by acute doses of both drugs. Future studies in stress models of depression should be subjected to unravel the basis of the antidepressant effect of psychedelics.

## Figures and Tables

**Figure 1 ijms-23-06713-f001:**
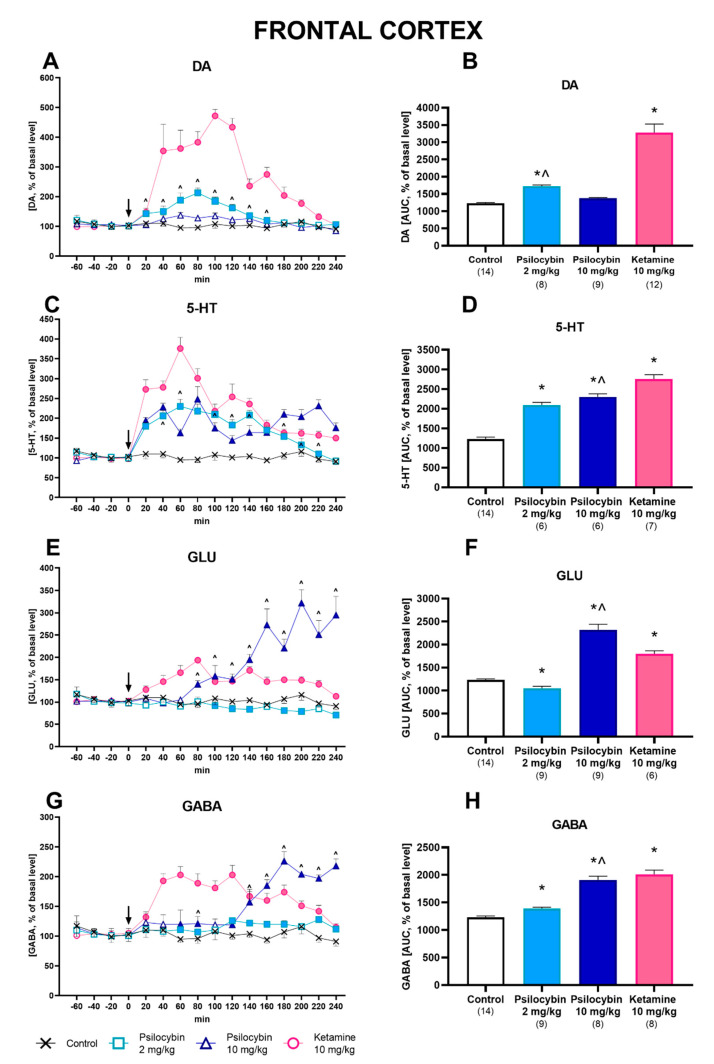
The time-course (**A**,**C**,**E**,**G**) and total (**B**,**D**,**F**,**H**) effect of psilocybin (2 and 10 mg/kg) and ketamine (10 mg/kg) on the dopamine (DA), serotonin (5-HT), glutamate (GLU) and γ-aminobutyric acid (GABA) extracellular levels in the rat frontal cortex. The total effect is calculated as an area under the concentration-time curve (AUC) and expressed as a percentage of the basal level. Values are the mean ± SEM (n is given under the name of the group). The drug injection is indicated with an arrow. Filled symbols or * show statistical differences (*p* < 0.001) between control and drug treatment groups; ^ *p* < 0.001 show differences between psilocybin 2 and 10 mg/kg groups as estimated by repeated measures ANOVA (time-course) or one-way ANOVA (total effect) followed by Tukey’s post hoc test.

**Figure 2 ijms-23-06713-f002:**
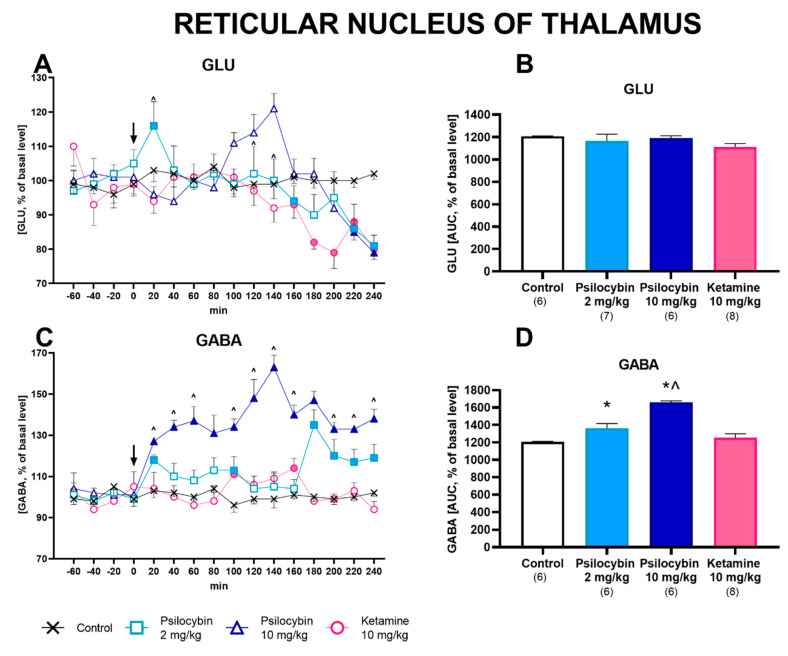
The time-course (**A**,**C**) and total (**B**,**D**) effect of psilocybin (2 and 10 mg/kg) and ketamine (10 mg/kg) on the glutamate (GLU) and γ-aminobutyric acid (GABA) extracellular levels in the rat reticular nucleus of the thalamus. The total effect is calculated as an area under the concentration-time curve (AUC) and expressed as a percentage of the basal level. Values are the mean ± SEM (n is given under the name of the group). The drug injection is indicated with an arrow. Filled symbols or * show statistical differences (*p* < 0.03–0.001) between control and drug treatment groups; ^ *p* < 0.005–0.001 show differences between psilocybin 2 and 10 mg/kg groups as estimated by repeated measures ANOVA (time-course) or one-way ANOVA (total effect) followed by Tukey’s post hoc test.

**Figure 3 ijms-23-06713-f003:**
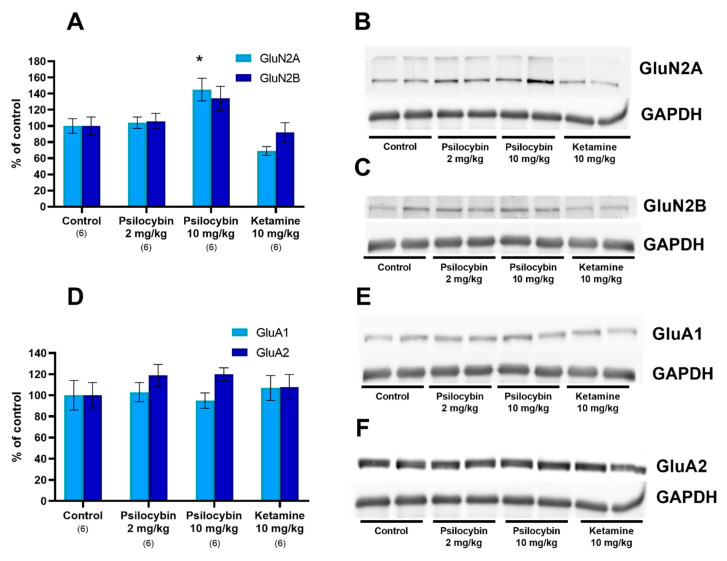
Levels of NMDA receptor subunits (GluN2A and GluN2B; (**A**) and AMPA receptor subunits (GluA1 and GluA2; (**D**) in the rat frontal cortex estimated 24 h after psilocybin (2 and 10 mg/kg) or ketamine (10 mg/kg) administration. The data are shown as percentages of the levels of the appropriate control groups. Each data point represents the mean ± SEM (n is given under the name of the group). Only in group GluN2A are data given in duplicates. * *p* < 0.05 vs. appropriate control group (one-way ANOVA followed by Tukey’s post hoc test). Examples of photomicrographs of the immunoblots using GluN2A and GAPDH antibodies (**B**), GluN2B and GAPDH antibodies (**C**), GluA1 and GAPDH antibodies (**E**) and GluA2 and GAPDH antibodies (**F**).

**Figure 4 ijms-23-06713-f004:**
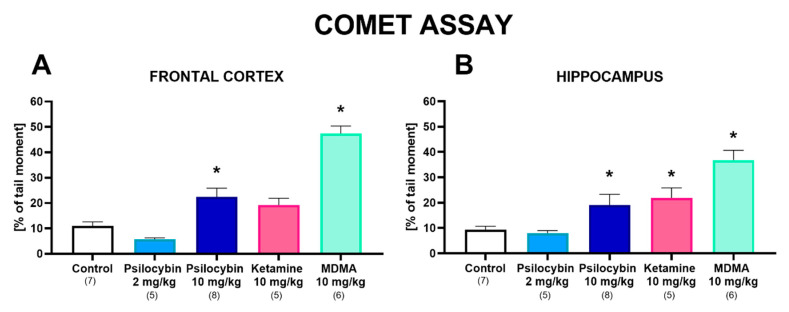
The effect of psilocybin (2 and 10 mg/kg), ketamine (10 mg/kg) and MDMA (10 mg/kg) on the oxidative damage of DNA in nuclei of the rat frontal cortex (**A**) and hippocampus (**B**) in the comet assay estimated 7 days after treatment. Data are the mean ± SEM (n is given under the name of the group) and represent tail moment shown as the product of the tail length and the fraction of total DNA in the tail. * *p* < 0.001–0.05 compared to the control (one-way ANOVA followed by Tukey’s post hoc test).

**Figure 5 ijms-23-06713-f005:**
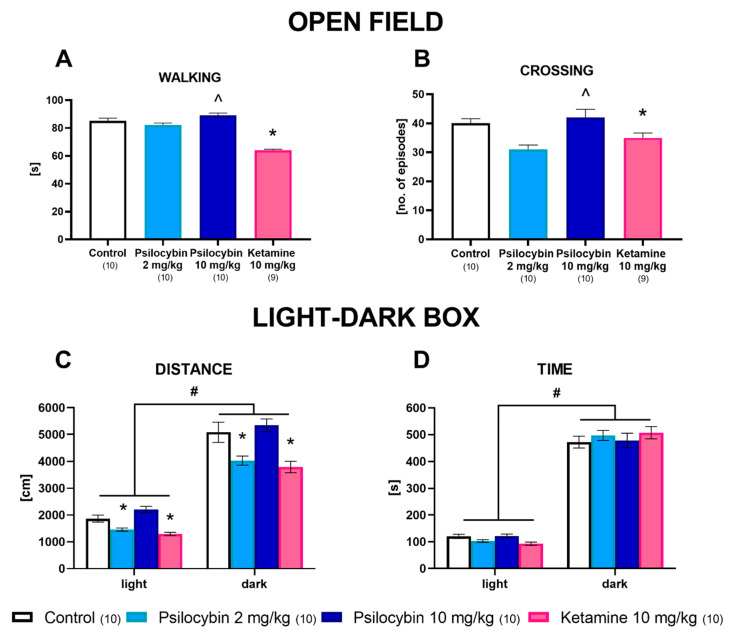
The effect of psilocybin (2 and 10 mg/kg) and ketamine (10 mg/kg) on locomotor behavior in the open field test (OF) and on activity of rats in the light–dark box (LDB) test 24 h after administration. The time spent on walking (**A**), the number of crossing episodes **(B**) in the OF test and ambulatory distance (**C**) and time spent (**D**) in the light and dark zone in the LDB test. Values are the mean ± SEM (n is given under or next to the name of the group). * *p* < 0.001–0.05 compared to the control; ^ *p* < 0.05 psilocybin 2 mg/kg compared to psilocybin 10 mg/kg; # *p* < 0.001 light vs. dark (one-way ANOVA or two-way ANOVA followed by Tukey’s post hoc test).

**Figure 6 ijms-23-06713-f006:**
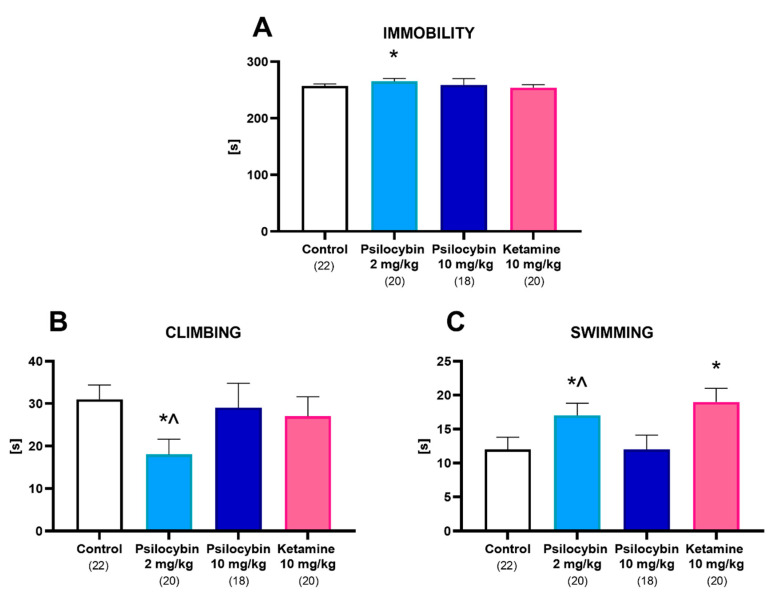
The effect of psilocybin (2 and 10 mg/kg) and ketamine (10 mg/kg) on the immobility (**A**), climbing (**B**) or swimming (**C**) time of rats in the forced swim test (FST) estimated 24 h after administration. Data are the mean ± SEM (n is given under the name of the group). * *p* < 0.001–0.01 compared to the control; ^ *p* < 0.05 psilocybin 2 mg/kg compared to psilocybin 10 mg/kg (one-way ANOVA followed by Tukey’s post hoc test).

## Data Availability

All data is contained within the article and Appendix A.

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
