# Peer review of "Effect of Psilocybin and Ketamine on Brain Neurotransmitters, Glutamate Receptors, DNA and Rat Behavior"

_ijms, 2022, doi:10.3390/ijms23126713_

Round 1

Reviewer 1 Report

The manuscript entitled “Effect of psilocybin and ketamine on brain neurotransmitters, glutamate receptors, DNA and rat behavior” by Wojtas and coworkers is focusing the research topic of great importance and actuality. The manuscript is well organized and written. It brings a new insight into a still controversial field. I find it very useful for the clarification of the clinical strategy, as well as for a better understanding of mechanisms underlying intriguing responses to drugs included in therapeutic protocols prior to the completion of preclinical investigations. Although I’ve enjoyed reading the manuscript, I have some remarks that should be addressed:

The reference list is missing.

What software for behavioral analysis was employed?

Insert the number of animals per analysis (parameter) per group and presented it on the x-axis below the name of the group.

Author Response

Dear Reviewer,

Thank you for your review. In reference to your remarks, below please find specific responses:

  1. The reference list is missing.
  • The references list seems to have been removed during the submission process. In the revised version it is added again.
  1. What software for behavioral analysis was employed?
  • Only light/dark box test was performed with the use of behavioral analysis software, namely: FCS Fear Conditioning System Version 06.03 (26.10.2007) Copyright HASOMED GmbH, Magdeburg a Member of the TSE-Group. The other behavioral tests were done manually by double-blind observers.
  1. Insert the number of animals per analysis (parameter) per group and presented it on the x-axis below the name of the group.
  • The number of animals per experimental group was added on the „x” axis below the name of each group.

Thank you again for time spent on reading our manuscript.

Reviewer 2 Report

This article is devoted to the study of the effect of ketamine and psilocybin on neurotransmitters and receptors. The relevance of this area is beyond doubt, since psychedelic substances are actively studied as therapeutic substances, and the effect of these substances on receptors and neurotransmitters is an important characteristic. The authors have done significant work in this direction. The article may be of interest to specialists in medicinal chemistry, psychopharmacology, etc. There are very important points that need to be finalized before this article can be recommended for publication:

1. Technical:

1.1 The article must be clearly formatted according to the template of this journal.

1.2 The article contains references to the literature, but there is no list of references at the end of the article. From this it is not clear what and with what the authors compare. Also, from this it is impossible to understand the completeness of the introduction, etc.

1.3. In biochemical studies on animals and humans, there must be an item "Declaration", in particular "Ethical norms and principles". It is important.

2. Actual items:

2.1 In the introduction, state more clearly the purpose of this study.

2.2 The conclusions should be substantially expanded.

2.3 Regarding the statistical processing of the obtained results, it is desirable to quote: 10.17516/1998-2836-0178

2.4 To assess the reliability of the data obtained, it is necessary to clarify in how many parallels the experiment was carried out. Did the authors do experiments on reproducibility?

2.5 Further consideration is possible in the presence of a list of references, since it is not clear with which work the authors are comparing their data.

Author Response

Dear Reviewer,

Thank you for your review. Below are our responses to your comments.

Ad. 1. Technical:

1.1.The article must be clearly formatted according to the template of this journal.

The revised article is to be submitted according to the template of the journal.

1.2. The article contains references to the literature, but there is no list of references at the end of the article. From this it is not clear what and with what the authors compare. Also, from this it is impossible to understand the completeness of the introduction, etc.

The references list seems to have been removed during the submission process. In the revised version it is added again.

1.3. In biochemical studies on animals and humans, there must be an item "Declaration", in particular "Ethical norms and principles". It is important.

„Ethical norms” are added to the main text (again, this part of manuscript placed in „back matter” was also submitted)

Ad. 2. Actual items:

2.1. In the introduction, state more clearly the purpose of this study.

The purpose of the study was stated more clearly and an amended, current version of the Introduction is given below.

The mechanisms underlying the antidepressive properties of both ketamine and psilocybin are still not fully understood. The detailed comparison studies should be conducted to evaluate their mutual properties. In addition, their safety and long-lasting effects should be carefully explored if they are to be used as a standard treatment for mood disorders, not only as a curiosity applied in experimental therapies. To address those questions, we have examined the effect of a single administration of both ketamine and psilocybin on the extracellular levels of dopamine (DA), serotonin (5-HT), glutamate and -aminobutyric acid (GABA) in the rat frontal cortex and reticular nucleus of the thalamus, using in vivo microdialysis. The potential genotoxic effect in the frontal cortex and hippocampus was measured with the comet assay. Moreover, western blot analysis was performed to measure the effect of investigated drugs on selected protein levels in the rat frontal cortex. Finally, an open field test, light-dark box test and forced swim test were conducted to evaluate the effect on selected aspects of rat behavior. To distinguish long-lasting from acute, psychoactive effects, all the experiments with the exception of microdialysis were performed at least 24 hours after administration of the chosen drug. The effect of psilocybin and ketamine on neurotransmitters’ release was never studied in detail, but by direct comparison of their impact on neuronal network may explain mechanism of psychedelics in the brain. An identification of possible undesirable side effects of psilocybin and ketamine such as genotoxicity, behavioral disturbances and possible adaptive changes in glutamate receptors is of great importance. Our findings may be helpful in the definition of therapeutic capabilities of psychedelics.

2.2. The conclusions should be substantially expanded.

The conclusions were expanded to the current form below.

In conclusion, our results indicate that both psilocybin and ketamine exert a profound effect on thalamo-cortical neurotransmission. It seems likely that psilocybin and ketamine act  on various molecular targets. Psilocybin activates 5-HT2A receptors; ketamine blocks subsets of NMDA receptors on GABA interneurons, which in turn disinhibits pyramidal cells. Both mechanisms result in facilitation of glutamate release, which exerts stimulatory effect on dopaminergic VTA cells or serotonergic dorsal raphe neurons. As a consequence, this leads to an increase of DA and 5-HT levels. However, the modulatory role of 5-HT1A and 5-HT2C receptors in psilocybin effect cannot be excluded. Our findings also add neurochemical evidence that GABA neurons in the reticular nucleus of thalamus underlie the mechanism regulating the sensory information provided to the cortex by psilocybin. The increase in glutamate extracellular level in the frontal cortex after acute doses of psilocybin seems to correspond with changes in NMDA receptor subunit GluN2A level. The DNA damage produced by higher doses of psilocybin and ketamine may result from glutamate-induced excitotoxicity and oxidative stress. The changes observed in the level of neurotransmitters do not translate into rat behavior tested 24 h after administration. No effect on anxiety and reduction of immobility may result from adaptative mechanisms triggered by acute doses of both drugs. Future studies in stress models of depression should be subjected to unravel the basis of  antidepressant effect of psychedelics.

2.3. Regarding the statistical processing of the obtained results, it is desirable to quote: 10.17516/1998-2836-0178

Data analysis was conducted in line with accepted statistical methods, using universally used, professional and licensed computer software, i.e.  STATISTICA v.10 StatSoft Inc. 1984-2011 (USA) and GraphPad Prism v.5.00 GraphPad Software Inc. (USA). In our view, the suggested numerical optimalization used in the quoted publication is pointless. Moreover, the publication suggested by the reviewer is in Russian, not English (which is the standard scientific practice).

2.4. To assess the reliability of the data obtained, it is necessary to clarify in how many parallels the experiment was carried out. Did the authors do experiments on reproducibility?

Excluding western blot experiments, all other data were conducted on single animals and the number of animals is presented in figures and statistical analysis. Only data related to GluN2A subunits were replicated.

2.5. Further consideration is possible in the presence of a list of references, since it is not clear with which work the authors are comparing their data.

The references list seems to have been removed during the submission process. In the revised version it is added again.

Round 2

Reviewer 2 Report

The authors have significantly improved the quality of the publication. Now it can be accepted for publication.

Author Response

Dear Reviewer,

thank you for your positive opinion.

Best regards,

Krystyna Gołembiowska